# Contrastive Graph Distribution Alignment for Partially View-Aligned Clustering

Xibiao Wang*
Shantou University
Shantou, Guangdong, China
w1574485261@gmail.com

Hang Gao*
Jilin University
Changchun, Jilin, China
gaohang23@mails.jlu.edu.cn

Xindian Wei
City University of Hong Kong
Hong Kong, China
xindiawei2-c@my.cityu.edu.hk

Liang Peng
Shantou University
Shantou, Guangdong, China
23lpeng@stu.edu.cn

Rui Li
Shantou University
Shantou, Guangdong, China
rli@stu.edu.cn

Cheng Liu†
Shantou University
Shantou, Guangdong, China
chengliu10@gmail.com

Si Wu
South China University of Technology
Guangzhou, Guangdong, China
cswusi@scut.edu.cn

Hau-San Wong
City University of Hong Kong
Hong Kong, China
cshswong@cityu.edu.hk

## Abstract

Partially View-aligned Clustering (PVC) presents a challenge as it requires a comprehensive exploration of complementary and consistent information in the presence of partial alignment of view data. Existing PVC methods typically learn view correspondence based on latent features that are expected to contain common semantic information. However, latent features obtained from heterogeneous spaces, along with the enforcement of alignment into the same feature dimension, can introduce cross-view discrepancies. In particular, partially view-aligned data lacks sufficient shared correspondences for the critical common semantic feature learning, resulting in inaccuracies in establishing meaningful correspondences between latent features across different views. While feature representations may differ across views, instance relationships within each view could potentially encode consistent common semantics across views. Motivated by this, our aim is to learn view correspondence based on graph distribution metrics that capture semantic view-invariant instance relationships. To achieve this, we utilize similarity graphs to depict instance relationships and learn view correspondence by aligning semantic similarity graphs through optimal transport with graph distribution. This facilitates the precise learning of view alignments, even in the presence of heterogeneous view-specific feature distortions. Furthermore, leveraging well-established cross-view correspondence, we introduce a cross-view contrastive learning to learn semantic features by

exploiting consistency information. The resulting meaningful semantic features effectively isolate shared latent patterns, avoiding the inclusion of irrelevant private information. We conduct extensive experiments on several real datasets, demonstrating the effectiveness of our proposed method for the PVC task.

## CCS Concepts

• **Theory of computation → Unsupervised learning and clustering**; • **Computing methodologies → Cluster analysis**.

## Keywords

Partially View-Aligned Clustering, Multi-View Clustering

### ACM Reference Format:

Xibiao Wang, Hang Gao, Xindian Wei, Liang Peng, Rui Li, Cheng Liu, Si Wu, and Hau-San Wong. 2024. Contrastive Graph Distribution Alignment for Partially View-Aligned Clustering. In *Proceedings of the 32nd ACM International Conference on Multimedia (MM '24), October 28-November 1, 2024, Melbourne, VIC, Australia.* ACM, New York, NY, USA, 10 pages. https://doi.org/10.1145/3664647.3681048

## 1 Introduction

In multimedia applications, datasets frequently incorporate diverse feature representations, including images, text, and videos for each sample. These various representations are collectively known as multi-view data. Multi-View Clustering (MVC) is aimed at improving performance by leveraging the inherent consistency and complementary attributes within multi-view data [9, 21, 28, 34, 55]. While various approaches have empirically demonstrated success in addressing the MVC task with different strategies [4, 13, 25, 35, 39, 40, 43, 45], the effectiveness of existing MVC methods relies on the idealized assumption that every view is perfectly aligned. In reality, this assumption can be easily violated during imperfect data collection, where only a portion of samples exhibit alignment across views. This leads to the problem of partially view-aligned clustering (PVC), as shown in Fig. 1. There is a critical need for models that can precisely learn view correspondences, such that

---

*Both authors contributed equally to this research.
†Corresponding author.

*MM '24, October 28-November 1, 2024, Melbourne, VIC, Australia*
© 2024 Copyright held by the owner/author(s). Publication rights licensed to ACM.
ACM ISBN 979-8-4007-0686-8/24/10
https://doi.org/10.1145/3664647.3681048

consistent and complementary information can be exploited within partially aligned multi-view data.

Recently, several methods have been proposed to tackle the challenges of PVC [8, 12, 38, 50]. For example, Huang *et al.* introduced a differentiable surrogate of the Hungarian algorithm within a deep learning framework to establish correspondences for view alignment between two views [8]. In addition, Yang *et al.* developed a view alignment representation learning model that learns view correspondences employing a noise-robust contrastive loss [48]. While existing PVC methods have achieved remarkable success, they primarily rely on the Euclidean distances between latent feature representations in pairs of views to establish view correspondences. However, these approaches assume that the learned latent features should encode consistent common semantics across views. In practice, latent features are often learned from different heterogeneous feature spaces, introducing the possibility of inconsistent view-private information. Forcing multiple views into the same feature space dimension can exacerbate cross-view discrepancies. Furthermore, when only a subset of samples exhibits across-view alignment, there may be insufficient shared information for the critical common semantic feature learning required for accurate correspondence. Consequently, directly measuring view differences and correspondences from inconsistent latent spaces can introduce discrepancies, hindering the accurate inference of cross-view sample correspondences based solely on latent representations.

While feature representations may exhibit divergence across views, the instance relationships within each view often maintain consistency and are considered to encode more reliable view-invariant information. Consequently, effectively capturing such semantic view-invariant instance relationships can provide a promising approach for learning view correspondence. Inspired by this insight, we introduce graph distribution-matched contrastive learning for partially view-aligned clustering. This method aims to learn view correspondence utilizing graph distribution metrics that accurately capture semantic view-invariant instance relationships. Moreover, it leverages well-learned view-alignment to exploit consistency and complementary information, thereby effectively learning meaningful semantic features. Specifically, as depicted in Fig. 1, our method estimates similarity graphs on latent features to capture instance relationships for each view. Notably, different views tend to produce similar graphs even when they are derived from view-specific latent features. Leveraging these semantic-invariant structural information across views, we employ optimal transport with graph distribution to precisely establish view alignment by matching these structural graph similarities. To further enhance clustering performance, it is crucial to leverage consistent information across views by exploiting learned correspondences. As discussed previously, latent spaces may contain irrelevant view-specific signals, and directly applying consistency learning to these latent spaces may result in the dominance of such signals over meaningful semantics. To address this challenge, we propose a feature extraction strategy that focuses on learning semantic features associated with similarity graphs for each view. Additionally, we introduce a contrastive cross-view feature learning approach in conjunction with semantic feature learning to guide the exploitation of consistency based on the learned alignment. This contrastive cross-view feature learning approach effectively isolates shared latent

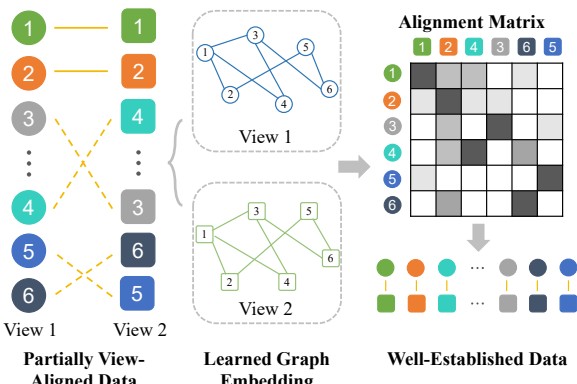

**Figure 1: This example illustrates the partially view-aligned problem and our proposed solution. Different shapes represent instances from different views. Solid lines indicate a small portion of given view correspondence between samples. However, most correspondence relationships (dashed lines) are missing, reflecting partial alignment. Our method learns the view-alignment matrix to capture complete correspondence. We achieve this by measuring differences between learned graphs from the two views via correspondence learning.**

patterns and mitigates the capture of irrelevant private information, by leveraging the view-alignment learning. As a result, the learned meaningful semantic features optimally harness multi-view consistency, leading to improved multi-view clustering performance.

The main contributions of this work could be summarized as follows:

- Diverging significantly from the conventional reliance on latent feature spaces with Euclidean distances, we introduce a novel perspective by leveraging similarity graphs to capture semantic-invariant structural information across views and propose a graph distribution-matched approach, precisely establishing view alignment through effectively matching of these structural graph similarities.
- To leverage the well-established cross-view correspondence, we introduce a cross-view contrastive loss along with learning semantic features to exploit consistency information. The acquired meaningful semantic features effectively isolate shared latent patterns, avoiding the capture of irrelevant private information.
- A comprehensive model analysis study illustrates the detailed effectiveness of our approach. The results confirm that the proposed approach effectively addresses the challenge of learning precise view correspondences in partially aligned multi-view data.

## 2 Related Work

Multi-view clustering (MVC) aims to leverage the consistency and complementarity information present in diverse views, leading to the development of various methodologies based on different assumptions [6, 19, 27, 32, 48]. For instance, Li *et al.* proposed the

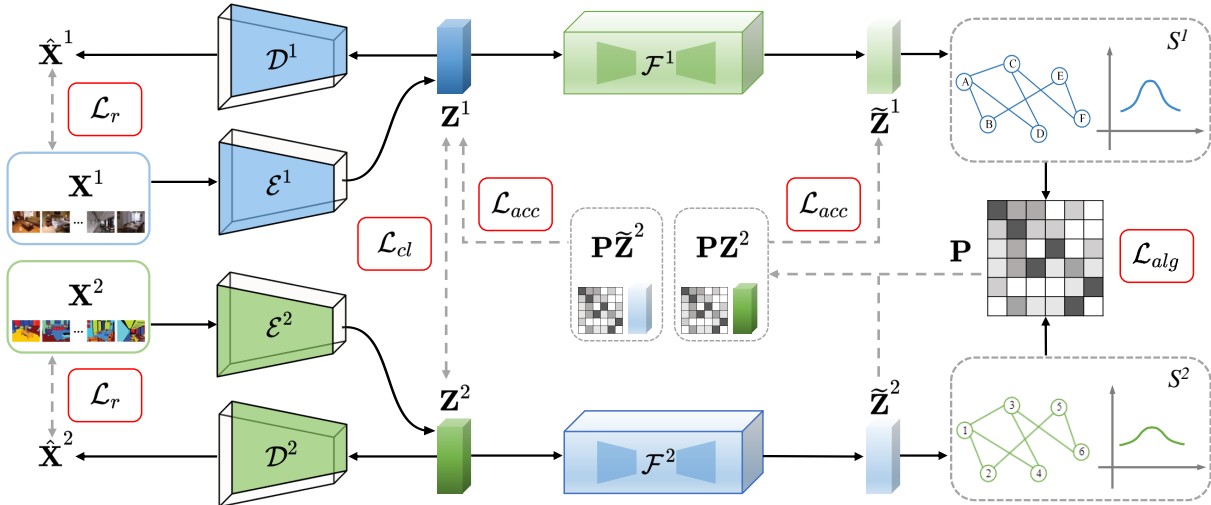

**Figure 2: The architecture depicts our model processing partially view-aligned multi-view data $\{X^v\}_{v=1}^m$. Each view passes through view-specific autoencoders to learn latent features $\{Z^v\}_{v=1}^m$, regularized by a contrastive loss $\mathcal{L}_{cl}$ over known sample correspondences. Leveraging the cross-view alignment matrix P from our matching module, we generate semantic features $\{\widetilde{Z}^v\}_{v=1}^m$ (via $\mathcal{F}$) anchored by an aligned contrastive loss $\mathcal{L}_{acc}$. This extracts meaningful signals to improve representation learning. The alignment module matches structural similarity graphs and thus learn precise view correspondences based on distribution distance metrics invariant to view semantics.**

Deep Adversarial MVC method, employing adversarial training to enhance the understanding of intrinsic structures within multi-view data [13]. Meanwhile, Yang *et al.* developed a collaborative MVC approach, employing inter-view collaborative learning to harness complementary information across multiple clustering tasks [49]. Yan *et al.* introduced a global and cross-view feature aggregation method, known as GCFagg [46], for learning global similarity relationships among samples. This method utilizes contrastive learning to align view-specific representations with a consensus representation.

While these approaches have shown promise in multi-view clustering, they often rely on assumptions of complete and perfectly aligned views. In real-world scenarios, these assumptions may be violated, leading to challenges in Incomplete Multi-View Clustering (IMVC) and Partially View-Aligned Clustering (PVC). IMVC assumes the potential absence of some instances in certain views within the dataset, and IMVC methods aim to address the challenge of missing views [15, 16, 18, 36, 42]. For example, Xu *et al.* employed generative adversarial networks (GANs) to recover missing data [44], and Liu *et al.* proposed a graph completion approach for inferring similarity graphs for missing views [17]. Additionally, Tang *et al.* introduced a unified framework with theoretical guarantees, aiming to simultaneously extract semantically consistent imputations and minimize the risk of clustering performance degradation caused by semantically inconsistent imputations [30].

In addition to MVC and IMVC, PVC focuses on establishing cross-view correspondences, with several approaches emerging to tackle this problem [8, 12, 38, 52, 54]. For instance, Huang *et al.* introduced a differentiable surrogate of the Hungarian algorithm within a deep learning framework to establish correspondences for

view alignment [8], and Yang *et al.* developed a view alignment representation learning model that learns view correspondences using a noise-robust contrastive loss [48]. Moreover, Yang *et al.* proposed a novel contrastive learning method, termed robust Multi-View Clustering with Incomplete Information (SURE), which addresses both IMVC and PVC challenges under a unified framework [47].

In contrast to existing PVC methods that rely on Euclidean distances between feature representations, this work introduces an optimal transport-based graph matching approach to address the challenges inherent in PVC. Optimal transport, a foundational tool for quantifying similarity between distributions, has been successfully applied in a variety of domains, demonstrating its versatility and effectiveness, including feature aggregation [22], object detection [5], and domain adaptation [11, 23]. By determining the most efficient redistribution of mass, optimal transport facilitates the assessment of distribution alignment. By integrating optimal transport into measuring view graph similarities for graph matching, the proposed approach constructs cross-view correspondences from a graph matching perspective, offering a novel viewpoint in the PVC domain.

## 3 Methodology

### 3.1 Problem Formulation

Consider a partially view-aligned dataset denoted as $\{\mathbf{X}^v\}_{v=1}^m$, consisting of $n$ instances across $m$ views. The dataset can be divided into two distinct subsets: aligned data and unaligned data. Subsequently, an index set $\Omega$ is introduced to represent the sample indices of the aligned data. The primary objective is to align the unaligned

data by leveraging the information from the aligned subset and improve the clustering performance by exploiting the complementary and consistency information with the well-established cross-view correspondence.

Autoencoder networks serve as foundational tools for transforming raw input features into latent representations, thereby facilitating unsupervised clustering. To initiate this process, we employ view-specific autoencoders to obtain latent feature representations for each view. Specifically, for each view, given the original data $\mathbf{X}^v$, we utilize an encoder network $\mathcal{E}^v$ to learn latent features: $\mathbf{Z}^v = \mathcal{E}^v(\mathbf{X}^v)$. Subsequently, a decoder network $\mathcal{D}^v$ is applied to reconstruct data based on these latent features: $\hat{\mathbf{X}}^v = \mathcal{D}^v(\mathbf{Z}^v)$. Here, $\hat{\mathbf{X}}^v$ represents the reconstruction of $\mathbf{X}^v$ for the $v$-th view. To preserve view-specific information within the latent representations during the reconstruction process, a reconstruction loss is introduced to measure the disparity between the original data and their corresponding reconstructions, and it can be expressed as:

$$\mathcal{L}_r = \sum_{v=1}^{m} \left\| \mathbf{X}^v - \hat{\mathbf{X}}^v \right\|_F^2 . \tag{1}$$

**Motivation.** MVC is capable of leveraging the inherent consistency and complementary information present in various views. However, in the context of PVC, only a subset of the data exhibits alignment across views. Therefore, a crucial step in the PVC task is to establish correspondence between pairs of views. For instance, when considering two views, the essence of clustering partially view-aligned data is to find a correspondence, often represented as a permutation matrix $\mathbf{P} \in \mathbb{R}^{n \times n}$, such that:

$$\mathbf{Z}^1 \sim \mathbf{P}\mathbf{Z}^2. \tag{2}$$

It describes the relationship between the latent representations in the two views. Fig. 2 illustrates the structure of the proposed method, which comprises two primary modules. The first module is focused on acquiring contrastive cross-view representation learning through the utilization of predicted correspondences $\mathbf{P}$, while the second module is dedicated to estimating view correspondences via optimal transport-based graph matching. These two modules can reinforce each other in the proposed model.

### 3.2 Contrastive Cross-View Feature Learning

We introduce a contrastive cross-view representation learning module to harness consistency and complementary information across views. Specifically, to leverage the consistency of feature information relying on the known correspondences of a subset of samples, we first adopt a contrastive learning approach aimed at maximizing mutual information across different views [14]. The formulation of this contrastive learning loss is as follows:

$$\mathcal{L}_{cl} = - \sum_{i \in \Omega} (I(\mathbf{Z}_\Omega^1, \mathbf{Z}_\Omega^2) + \alpha(H(\mathbf{Z}_\Omega^1) + H(\mathbf{Z}_\Omega^2))), \tag{3}$$

where $\mathbf{Z}_\Omega$ represents the subset of view-specific features derived from the known correspondences part of the samples. $I$ denotes mutual information, $H$ signifies information entropy, and the parameter $\alpha$ serves as an entropy regularization term. The term $I(\mathbf{Z}_\Omega^1, \mathbf{Z}_\Omega^2)$ encourages $\mathbf{Z}_\Omega^1$ and $\mathbf{Z}_\Omega^2$ to acquire greater view-consistent information, while $H(\mathbf{Z}_\Omega^1)$ and $H(\mathbf{Z}_\Omega^2)$ are employed to prevent the trivial

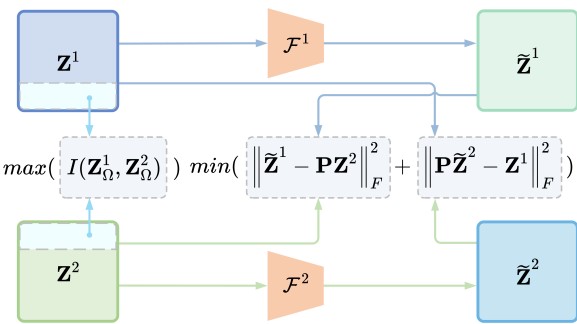

**Figure 3: Illustration of the proposed cross-view feature learning strategy. The latent feature representations for each view are obtained and constrained through contrastive learning applied to known correspondence. Additionally, both the semantic features and latent features are constrained by the aligned cross-view contrastive learning loss.**

scenario in which all samples exhibit uniform consistency. However, relying solely on contrastive learning with known aligned samples limits the ability to harness multi-view consistency in unaligned data. Moreover, strictly contrasting features may lead to the dominance of irrelevant view-private noises, which can distort the extraction of semantic consistency and overlook available complementary information.

To address this challenge, we aim to learn semantic features for each view to aid in discovering consistency information. ***However, without any label information, directly learning semantically meaningful features is challenging. As discussed, similarity graphs convey view-invariant semantic structures of instance relationships.*** Here, we introduce a feature extraction strategy to generate semantic features that leverages the learning of similarity graphs by connecting it with view alignment, as detailed in Section 3.3. Subsequently, we propose a cross-view contrastive loss to guide the exploitation of consistency based on the learned semantic features and view-alignment. Specifically, we employ modules $\mathcal{F}^v$ on the latent representations of each view $\mathbf{Z}^v$ for semantic feature learning:

$$\widetilde{\mathbf{Z}}^1 = \mathcal{F}^1(\mathbf{Z}^1), \tag{4}$$

$$\widetilde{\mathbf{Z}}^2 = \mathcal{F}^2(\mathbf{Z}^2). \tag{5}$$

The generated semantic feature $\widetilde{\mathbf{Z}}^1$ is expected to exhibit consistency with $\mathbf{Z}^2$, and vice versa for $\widetilde{\mathbf{Z}}^2$. Consequently, these generated semantic features can also serve as view-consistent information guiding feature learning. To align with the nature of our partially view-aligned dataset, we leverage the alignment matrix $\mathbf{P}$ to assist in learning the semantic and latent features with an aligned cross-view contrastive loss:

$$\mathcal{L}_{acc} = \left\| \widetilde{\mathbf{Z}}^1 - \mathbf{P}\mathbf{Z}^2 \right\|_F^2 + \left\| \mathbf{P}\widetilde{\mathbf{Z}}^2 - \mathbf{Z}^1 \right\|_F^2 . \tag{6}$$

As illustrated in Fig. 3, this cross-view contrastive loss facilitates the learning of both latent features and semantic features based on the learned alignment matrix $\mathbf{P}$. This module allows for referencing both latent features and semantic features, exploiting consistency information to enhance the learning process.

## 3.3 Cross-View Graph Alignment

The primary challenge in our proposed method lies in extracting view alignment information, specifically computing the permutation matrix $\mathbf{P}$, and integrating it with the learning of semantic features to guide cross-view contrastive learning. Existing approaches [8, 48] calculate the view-alignment $\mathbf{P}$ based on Euclidean distances between latent features across views, which may incorporate meaningless view-private information, leading to erroneous view correspondences during the learning process. Here, our method leverages similarity graphs to capture view-invariant semantic structures of instance relationships from a graph alignment perspective. However, directly applying the Euclidean distance between similarity graphs for view-alignment is not feasible, as it presents challenges in exploring the global structure among the graphs [26]. Instead, the Gromov-Wasserstein distance of graph structures and the Wasserstein distance of graph signals offer effective measures for quantifying the dissimilarity between two graphs. Inspired by [26], we consider employing the Wasserstein distance on the distributions derived from similarity graphs based on semantic features. Specifically, the relation graphs with the learned semantic feature for each view can be first estimated as:

$$\mathbf{S}_{ij}^{v} = \frac{\left\langle \widetilde{\mathbf{Z}}_i^v, \widetilde{\mathbf{Z}}_j^v \right\rangle}{\left\| \widetilde{\mathbf{Z}}_i^v \right\|_2 \left\| \widetilde{\mathbf{Z}}_j^v \right\|_2}, \tag{7}$$

where $\langle \cdot, \cdot \rangle$ represents the inner product of two vector. Subsequently, we define the graph distribution as a normal distribution with a mean of $\mathbf{0}$ and a covariance matrix of $\mathbf{L}^\dagger$, leveraging the similarity graph $\mathbf{S}$ and its Laplacian $\mathbf{L} = \mathbf{D} - \mathbf{S}$:

$$\mu = \mathcal{N}(0, \Sigma), \tag{8}$$

where $\Sigma = \mathbf{L}^\dagger$ is the covariance matrix of this distribution and $\dagger$ denotes the pseudoinverse operator. Furthermore, the graph distribution with the alignment matrix $\mathbf{P}$ can be further defined as:

$$\mu_{\mathbf{P}} = \mathcal{N}(0, \mathbf{P}\Sigma\mathbf{P}^\top). \tag{9}$$

Let $\mu^{\mathbf{S}^1}$ and $\mu_{\mathbf{P}}^{\mathbf{S}^2}$ denote the distributions of graphs on view 1 and alignment graphs on view 2, respectively. The Wasserstein distance $\mathcal{W}2^2(\mu^{\mathbf{S}^1}, \mu_{\mathbf{P}}^{\mathbf{S}^2})$ between these distributions quantifies the dissimilarity between two graphs, and optimizing this distance can facilitate the alignment process $\mathbf{P}$. In term of these definitions, we develop a variant of the graph matching formulation [26] to design the graph alignment module, aiming to find a permutation that minimizes the $L_2$-Wasserstein distance between the distributions of the graph and the alignment graph, and have the following formulation of:

$$\mathcal{L}_{alg} = tr\left(\Sigma_1\right) + tr\left(\Sigma_2\right) + tr\left(\sqrt{\Sigma_1^{\frac{1}{2}} \mathbf{P}\Sigma_2^{\frac{1}{2}} \mathbf{P}^\top \Sigma_1^{\frac{1}{2}}}\right), \tag{10}$$

where $\mathbf{P} = \left\{ \mathbf{P} \in \mathbb{R}^{n \times n} \mid \mathbf{P} \in [0, 1]; \mathbf{P}_\Omega = 1; \mathbf{1}^\top \mathbf{P} = \mathbf{1}; \mathbf{P}\mathbf{1} = \mathbf{1}; \mathbf{P}^\top \mathbf{P} = \mathbf{I} \right\}$ and $\mathbf{P}_\Omega = 1$ denotes the known correspondences, which are assigned as 1 in this alignment matrix. This optimization problem can be solved using stochastic gradient descent [26].

**How Does Our Model Exploit Consistency Information with Partially Alignment?** The optimal $\mathbf{P}$ facilitates the exploration of assignments aimed at matching two partially aligned views by assessing the similarity between graphs and enforcing shared structural characteristics (graph distribution) between the views. In addition, this alignment matrix serves as a guide for contrastive cross-view feature learning to learn the semantic features by addressing the challenge of partially aligned views. In turn, the learned semantic features that are associated with the graph structure help to accurately estimate reliable graph distributions, further enhancing the learning process of the view alignment matrix. By combining structural graph alignment and semantic feature learning, this collaborative learning strategy improves overall clustering performance, capturing the intrinsic relationships among diverse data representations.

## 3.4 Optimization

We employ an alternating optimization approach to iteratively update the parameters of the autoencoder and the alignment matrix $\mathbf{P}$. The training process can be outlined as the following steps:

- We adopt a pretraining strategy to initially optimize the entire network using data with known view correspondence. In the pretraining stage, the reconstruction loss $\mathcal{L}_r$ and contrastive loss $\mathcal{L}_{cl}$ are utilized to train the encoder and decoder, mapping the input data to appropriate latent space.
- In the subsequent epochs, unaligned data is included to perform representation learning and alignment in an iterative order. Specifically, the contrastive representation module is employed to learn the latent features and the semantic features. The view alignment module is trained on the similarity graphs with $\mathcal{L}_{alg}$ to obtain the alignment matrix $\mathbf{P}$. These two modules are referenced and learned in a cyclical manner.
- Finally, a common representation is obtained by concatenating the view-specific semantic features, followed by the application of the K-means clustering algorithm to derive the final clustering results.

An overview of the optimization algorithm for the proposed method can be found in the supplementary file.

## 4 Experiments

In this section, we conduct experiments on several widely-used multi-view datasets to validate the effectiveness of our method. We compare its performance with several state-of-the-art methods and provide a comprehensive analysis of the model.

### 4.1 Experimental Settings

**Datasets:** six widely-used multi-view datasets are adopted to evaluate the effectiveness of our model, including **HandWritten [41], Scene-15 [8], BDGP [3], Caltech101-7 [52], Caltech101-20 [52], Reuters-dim10 [48].** A brief summary of the information for these datasets is provided in Table 1.

**Competing methods:** in this experimental study, we compare our method with 12 standard multi-view clustering methods, including: CCA [31], KCCA [2], DCCA [1], DCCAE[37], MvC-DMF [53], SwMC[24], GMC [33], AE$^2$-NETs [51], LMVSC [10], SMVSC [29], OPMV [20], FastMICE [7] and 4 partially view-aligned multi-view clustering methods: MVC-UM [50], PVC [8], MvCLN [48], SURE

**Table 1: Statistics of the datasets.**

| Dataset | Samples | Classes | Features |
|---|---|---|---|
| **HandWritten** | 2,000 | 10 | {240,216} |
| **Scene-15** | 4,485 | 15 | {50,59} |
| **BDGP** | 2,500 | 5 | {1,750,79} |
| **Caltech101-7** | 1,474 | 7 | {1,984,512} |
| **Caltech101-20** | 2,386 | 20 | {1,984,512} |
| **Reuters-dim10** | 9,379 | 6 | {10,10} |

[47]. For all competing approaches, we report their experimental results from related papers or use the published source code and conduct experiments following the optimal parameter settings provided by the authors.

**Construction of partially aligned data:** to assess the performance in handling partially view-aligned data, we first generate such data from multi-view data. Specifically, to create $\gamma$ partially aligned multi-view data, we select $\gamma$ of samples from all views as fixed known correspondences. The remaining samples are randomly shuffled to generate the unaligned parts of this data.

**Evaluation metrics:** to evaluate the clustering performance, we adopt three widely used metrics: Accuracy (ACC), Normalized Mutual Information (NMI), and Adjusted Rand Index (ARI). Higher values of ACC, NMI, and ARI indicate better clustering performance.

## 4.2 Comparative Analysis

We evaluate our method by comparing with other competing approaches in two settings: partially aligned multi-view data where the alignment ratio is 50% and fully aligned multi-view data. Notably, methods such as PVC, MVC-UM, MvCLN, and SURE are specifically designed to handle partially aligned multi-view data. For other standard MVC approaches, to address the unaligned data, we employ PCA for dimensionality reduction, followed by the utilization of the Hungarian algorithm to establish the mapping relationship among samples. This transformation effectively converts the partially aligned multi-view data into the context of fully aligned multi-view clustering. Tables 2 and 3 present the clustering performance comparison on these benchmark datasets under partially and fully aligned multi-view settings, respectively. Based on these results, we have the following observations:

- Compared with most traditional MVC methods, which are unable to handle partially-aligned view data, PVC, MVC-UM, MvCLN, SURE and our method consistently demonstrate superior clustering results on that, outperforming other methods. This advantage can be attributed to the fact that these models initially consider the partially view-aligned problem. In contrast to using the Hungarian algorithm to construct mapping relationships, these methods are able to learn accurate view alignment, addressing the challenge of partially-aligned view data.
- Obviously, compared with other PVC methods, our method achieves the best results on most datasets. These findings demonstrate the effectiveness of our method in feature learning and its accuracy in constructing cross-view mapping relationships.

- Furthermore, our method achieves better and more competitive performance even when compared with traditional MVC methods at a fully aligned setting (without missing view correspondences). Especially, the results of our method on partially-aligned view data significantly surpass that of standard MVC methods on Caltech101-7 with fully aligned data. The results consistently demonstrate that our method achieves comparable and, in many cases, superior performance in this scenario. These findings provide additional validation for the effectiveness of the proposed feature learning framework.

## 4.3 Model Analysis

In this section, we conduct a comprehensive analysis of our method from various perspectives, encompassing ablation studies, an examination of the impact of different alignment ratios, and validation of the view alignment module.

*4.3.1 Ablation Studies.* We conduct an ablation study to assess the effectiveness of key components in our method: the cross-view contrastive learning and the view alignment learning. The impact of the loss functions $\mathcal{L}_{acc}$ and $\mathcal{L}_{cl}$ of the cross-view contrastive learning is comparatively evaluated on the Caltech101-7, Caltech101-20, and BDGP datasets. Additionally, we evaluate the view alignment module, $\mathcal{L}_{alg}$, by comparing it with the Hungarian algorithm (wherein the alignment matrix $\mathbf{P}$ is learned without the view alignment module). As illustrated in Table 4, we observe that the clustering performance improves with the inclusion of each model component. This consistent improvement validates that each component contributes significantly to enhancing clustering outcomes.

*4.3.2 **Does the model exhibit robustness to different view-alignment ratios?*** We conduct experiments to assess the robustness of our method under different data alignment ratios. Specifically, we vary the alignment ratio ($\gamma$) from 20% to 100% with a gap of 20% on the Caltech101-20 and Reuters data. These results are depicted in Fig. 4. From them, we observe that: the more available alignment data, the better the performance of all PVC methods become, and the proposed method shows a significant increase. Especially, when the alignment ratio is 80%, our results are already close to the results of fully aligned data. Our method consistently outperforms other competing methods across different alignment ratios of the data. Even with a low alignment ratio, our method achieves satisfactory results, verifying the robustness of the proposed method.

*4.3.3 **Is cross-view contrastive learning helpful?*** The effectiveness of cross-view feature learning can be assessed through the quality of learned semantic features, which directly influences clustering ability. In Figure 5, we visualize the learned semantic features of the proposed method and competing PVC methods (PVC, MvCLN, and SURE). The results demonstrate that our method successfully captures accurate cluster structures, forming more compact clusters compared to the competing methods. This observation indicates that our method produces discriminative semantic feature representations, enhancing its performance in the clustering task.

 

**Table 2: Comparison of clustering performance with a *partially* aligned setting on six benchmark datasets, with the best results highlighted in red and the second-best results in blue.**

| Methods | HandWritten | | | Scene-15 | | | BDGP | | | Caltech101-7 | | | Caltech101-20 | | | Reuters | | |
|---|---|---|---|---|---|---|---|---|---|---|---|---|---|---|---|---|---|---|
| | ACC | NMI | ARI | ACC | NMI | ARI | ACC | NMI | ARI | ACC | NMI | ARI | ACC | NMI | ARI | ACC | NMI | ARI |
| CCA | 55.90 | 46.48 | 36.91 | 32.73 | 32.24 | 18.80 | 63.00 | 34.28 | 31.89 | 32.50 | 15.61 | 6.53 | 22.13 | 29.34 | 10.35 | 43.44 | 19.10 | 12.36 |
| KCCA | 38.85 | 29.53 | 19.05 | 33.09 | 31.43 | 16.35 | 47.32 | 21.26 | 17.56 | 26.87 | 9.34 | 3.59 | 16.01 | 17.47 | 4.52 | 29.40 | 5.31 | 3.80 |
| DCCA | 36.10 | 40.69 | 25.75 | 34.27 | 36.55 | 18.83 | 56.60 | 25.88 | 22.60 | 30.46 | 10.39 | 5.67 | 21.50 | 25.39 | 8.90 | 33.10 | 7.36 | 4.96 |
| DCCAE | 43.55 | 48.13 | 34.18 | 33.62 | 36.56 | 18.54 | 59.20 | 28.29 | 25.57 | 47.42 | 32.22 | 20.91 | 31.56 | 40.38 | 17.47 | 31.50 | 7.00 | 5.06 |
| MvC-DMF | 51.37 | 41.99 | 30.32 | 28.49 | 24.31 | 11.22 | 50.30 | 52.31 | 29.91 | 38.21 | 6.54 | 5.23 | 30.13 | 22.02 | 8.61 | 24.75 | 3.02 | 1.42 |
| SwMC | 21.50 | 14.78 | 7.86 | 31.03 | 30.39 | 12.94 | 36.75 | 8.64 | 7.23 | 46.12 | 9.31 | 2.08 | 33.82 | 1.16 | 0.39 | 27.41 | 0.04 | -0.01 |
| GMC | 43.50 | 51.29 | 30.50 | 11.08 | 4.51 | 0.10 | 49.92 | 28.40 | 23.55 | 53.73 | 17.61 | 8.64 | 39.10 | 20.51 | 7.86 | 26.98 | 4.50 | 0.46 |
| AE$^2$-NETs | 69.10 | 66.45 | 56.08 | 28.56 | 26.58 | 12.96 | 39.16 | 17.77 | 6.15 | 55.97 | 38.80 | 34.91 | 42.46 | 46.96 | 33.15 | 33.31 | 3.71 | 2.49 |
| LMVCS | 47.40 | 45.08 | 30.92 | 27.76 | 19.03 | 10.89 | 44.72 | 25.21 | 19.23 | 61.26 | 24.25 | 22.18 | 25.44 | 22.85 | 5.13 | 37.02 | 9.16 | 10.17 |
| SMVSC | 50.85 | 47.57 | 35.29 | 22.92 | 13.96 | 15.47 | 50.20 | 28.34 | 23.36 | 47.62 | 18.72 | 15.01 | 35.16 | 27.78 | 21.94 | 38.13 | 11.40 | 25.42 |
| OPMV | 43.10 | 47.56 | 28.99 | 29.90 | 27.15 | 12.55 | 46.48 | 19.71 | 9.17 | 34.19 | 17.59 | 12.08 | 22.67 | 30.09 | 11.96 | 37.48 | 10.62 | 7.77 |
| FastMICE | 48.93 | 58.34 | 37.51 | 30.00 | 25.29 | 13.20 | 53.49 | 30.19 | 25.75 | 41.47 | 27.55 | 22.14 | 25.50 | 29.61 | 12.22 | 36.16 | 14.90 | 11.20 |
| PVC | 76.45 | 74.47 | 66.22 | 37.88 | 39.12 | 20.63 | 89.24 | 73.56 | 74.93 | 50.14 | 53.54 | 38.38 | 48.95 | 64.19 | 38.34 | 35.34 | 16.12 | 11.55 |
| MVC-UM | 71.45 | 69.16 | 60.47 | 25.70 | 27.70 | 11.54 | 46.68 | 21.88 | 8.81 | 55.50 | 45.32 | 37.38 | 43.25 | 60.14 | 32.30 | 36.87 | 13.96 | 15.16 |
| MvCLN | 64.55 | 62.29 | 49.32 | 38.53 | 39.90 | 24.26 | 73.04 | 46.15 | 44.28 | 45.52 | 50.34 | 36.87 | 46.19 | 56.69 | 41.43 | 50.63 | 32.69 | 26.77 |
| SURE | 77.31 | 72.42 | 63.01 | 38.67 | 40.00 | 22.53 | 79.29 | 57.95 | 55.87 | 41.00 | 45.98 | 26.79 | 53.44 | 59.30 | 41.90 | 51.01 | 32.11 | 25.76 |
| Ours | 83.16 | 79.24 | 73.01 | 41.63 | 42.05 | 22.57 | 90.74 | 77.40 | 78.84 | 92.33 | 83.32 | 94.69 | 76.02 | 71.18 | 89.27 | 54.66 | 35.93 | 29.90 |

**Table 3: Comparison of clustering performance with a *fully* aligned setting on six benchmark datasets, with the best results highlighted in red and the second-best results in blue.**

| Methods | HandWritten | | | Scene-15 | | | BDGP | | | Caltech101-7 | | | Caltech101-20 | | | Reuters | | |
|---|---|---|---|---|---|---|---|---|---|---|---|---|---|---|---|---|---|---|
| | ACC | NMI | ARI | ACC | NMI | ARI | ACC | NMI | ARI | ACC | NMI | ARI | ACC | NMI | ARI | ACC | NMI | ARI |
| CCA | 71.10 | 70.15 | 60.85 | 36.37 | 36.91 | 19.82 | 81.52 | 62.53 | 61.45 | 58.14 | 62.53 | 48.80 | 47.07 | 59.56 | 38.82 | 44.01 | 21.96 | 15.20 |
| KCCA | 38.40 | 28.75 | 17.15 | 37.93 | 37.42 | 21.38 | 89.44 | 80.49 | 76.24 | 49.80 | 36.94 | 50.31 | 43.71 | 59.39 | 35.68 | 52.23 | 22.17 | 19.20 |
| DCCA | 71.75 | 68.03 | 59.54 | 36.61 | 39.20 | 21.03 | 96.72 | 91.58 | 91.56 | 47.69 | 54.23 | 37.13 | 41.70 | 61.07 | 34.76 | 52.30 | 34.02 | 24.57 |
| DCCAE | 34.15 | 35.00 | 20.34 | 34.58 | 39.01 | 19.65 | 93.12 | 90.37 | 85.73 | 52.29 | 57.19 | 45.80 | 41.32 | 56.03 | 30.44 | 43.81 | 34.73 | 19.08 |
| MvC-DMF | 70.04 | 68.18 | 58.09 | 30.99 | 31.35 | 15.68 | 35.26 | 11.47 | 4.98 | 41.56 | 18.44 | 5.23 | 50.13 | 50.90 | 31.92 | 24.85 | 3.05 | 1.88 |
| SwMC | 25.25 | 21.39 | 10.02 | 33.89 | 32.98 | 11.78 | 28.32 | 9.90 | 3.59 | 59.16 | 16.31 | 27.59 | 35.88 | 16.56 | 10.11 | 27.45 | 0.09 | -0.01 |
| GMC | 67.25 | 80.62 | 65.78 | 24.97 | 29.56 | 4.03 | 61.24 | 45.84 | 36.23 | 78.83 | 64.88 | 58.48 | 50.67 | 49.43 | 17.25 | 30.88 | 8.95 | 0.57 |
| AE$^2$-NETs | 79.40 | 76.27 | 70.13 | 37.17 | 40.47 | 22.24 | 57.52 | 45.83 | 16.83 | 61.26 | 45.75 | 42.58 | 40.32 | 53.20 | 31.57 | 41.13 | 8.24 | 7.57 |
| LMVCS | 69.45 | 70.22 | 58.22 | 35.12 | 35.58 | 18.92 | 44.72 | 25.21 | 19.23 | 66.28 | 51.65 | 42.84 | 45.43 | 64.76 | 37.64 | 45.93 | 17.68 | 17.19 |
| SMVSC | 70.05 | 64.19 | 52.49 | 36.76 | 35.33 | 25.98 | 61.48 | 45.52 | 36.58 | 68.86 | 61.52 | 52.73 | 62.28 | 58.46 | 57.06 | 42.52 | 18.91 | 33.18 |
| OPMV | 66.80 | 68.07 | 56.24 | 37.26 | 40.40 | 21.55 | 64.32 | 47.09 | 24.47 | 49.59 | 53.06 | 41.58 | 46.02 | 60.00 | 35.80 | 45.68 | 21.75 | 15.10 |
| FastMICE | 79.99 | 80.21 | 73.40 | 42.64 | 41.32 | 24.51 | 74.18 | 61.18 | 57.70 | 57.66 | 53.78 | 45.98 | 47.02 | 60.21 | 38.46 | 39.23 | 20.48 | 14.94 |
| PVC | 62.70 | 60.97 | 45.56 | 32.31 | 34.24 | 15.88 | 96.28 | 91.22 | 91.08 | 52.44 | 61.64 | 45.77 | 50.13 | 67.55 | 39.74 | 38.54 | 18.92 | 12.37 |
| MVC-UM | 84.25 | 73.65 | 68.60 | 35.45 | 37.02 | 19.56 | 60.92 | 48.33 | 19.20 | 56.17 | 58.84 | 35.17 | 43.50 | 61.26 | 33.05 | 36.90 | 13.58 | 14.74 |
| MvCLN | 81.23 | 73.55 | 66.17 | 42.46 | 40.05 | 23.67 | 77.59 | 55.68 | 53.65 | 45.64 | 47.72 | 30.86 | 40.93 | 54.64 | 32.84 | 57.72 | 33.89 | 28.97 |
| SURE | 73.76 | 71.98 | 62.77 | 41.59 | 39.46 | 22.83 | 93.39 | 85.37 | 85.02 | 42.74 | 45.08 | 29.00 | 48.98 | 58.56 | 38.65 | 56.89 | 32.74 | 28.82 |
| Ours | 87.65 | 80.00 | 75.71 | 41.49 | 42.98 | 23.64 | 98.04 | 94.07 | 95.19 | 87.45 | 78.50 | 92.42 | 82.10 | 78.29 | 93.38 | 57.51 | 37.78 | 32.89 |

**Table 4: Clustering results of different loss combinations. Among them, when $\mathcal{L}_{alg}$ is not added, we use the Hungarian algorithm for alignment.**

| $\mathcal{L}_{alg}$ | $\mathcal{L}_{acc}$ | $\mathcal{L}_{cl}$ | Caltech101-7 | | | Caltech101-20 | | | BDGP | | |
|---|---|---|---|---|---|---|---|---|---|---|---|
| | | | ACC | NMI | ARI | ACC | NMI | ARI | ACC | NMI | ARI |
| ✓ | | | 61.18 | 18.29 | 16.57 | 59.33 | 48.80 | 43.53 | 38.51 | 15.10 | 10.50 |
| ✓ | ✓ | | 73.49 | 59.12 | 69.49 | 63.44 | 56.60 | 56.55 | 44.72 | 29.14 | 17.04 |
| ✓ | | ✓ | 89.96 | 82.85 | 94.08 | 73.89 | 70.34 | 89.03 | 87.49 | 69.10 | 71.99 |
| ✓ | ✓ | ✓ | 92.33 | 83.32 | 94.69 | 76.01 | 71.18 | 89.27 | 90.74 | 77.40 | 78.84 |
| | | | 61.19 | 18.50 | 18.40 | 50.20 | 38.04 | 26.26 | 38.25 | 9.48 | 8.90 |
| | ✓ | | 70.42 | 56.13 | 66.37 | 59.87 | 52.91 | 46.33 | 43.62 | 20.32 | 7.38 |
| | | ✓ | 79.81 | 71.10 | 73.88 | 67.98 | 61.57 | 76.90 | 62.39 | 29.85 | 30.13 |
| | ✓ | ✓ | 88.97 | 78.35 | 89.80 | 68.63 | 61.79 | 77.59 | 89.75 | 75.21 | 76.74 |

*4.3.4 Can graph alignment accurately capture view correspondence?* The key of our proposed method lies in its capacity to learn correspondences for view-alignment. To further validate the effectiveness of the alignment module, we assess the quality of the learned view alignment matrix. Specifically, we use sankey diagram to compare the view-alignment obtained through the Hungarian algorithm, PVC, and our method on the BDGP and Caltech101-7 datasets, as illustrated in Figure. 6. As observed, the view correspondences learned by our method accurately identify view mappings across different clusters. The ground truth view alignment exhibits a one-to-one mapping between the two columns with the same cluster. In contrast, the view correspondences obtained through the Hungarian algorithm and PVC may exhibit inconsistencies

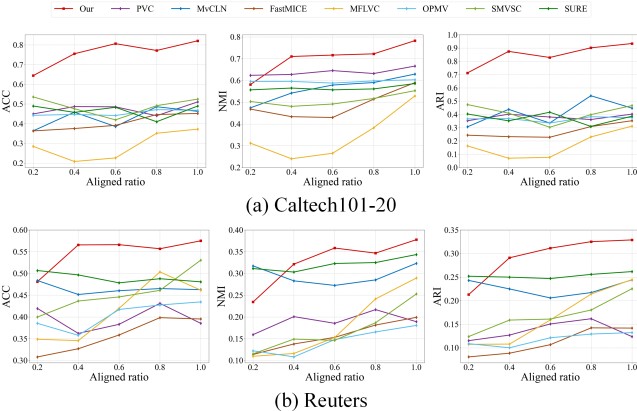

(a) Caltech101-20

(b) Reuters

**Figure 4: The clustering performance of our method is compared with several competing methods on the Caltech101-20 and Reuters datasets, considering varying alignment ratios. Please enlarge the figure for better visual results.**

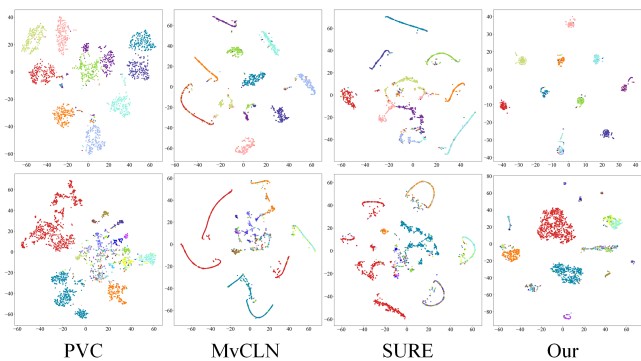

PVC          MvCLN          SURE          Our

**Figure 5: The t-SNE visualization of learned features of our method and competing PVC methods (PVC, MvCLN, and SURE) on the Handwritten and Caltech101-20 datasets. Please enlarge the figure for better visual results.**

with the ground truth view alignment. This result suggests that the alignment matrix learned by our method can precisely capture view-alignment information compared to the Hungarian algorithm and PVC approaches.

## 5 Conclusion

In this study, we propose a novel approach to tackle the practical challenge of Partially View-aligned Clustering (PVC). Unlike existing PVC methods that learn view correspondence based on latent features, our method leverages similarity graphs to capture the view-invariant semantic structures of instance relationships. We introduce graph distribution-matched contrastive learning for partially view-aligned clustering. This method effectively learns view correspondence by utilizing graph distribution metrics that capture semantic view-invariant instance relationships. The graph matching mechanism uncovers comprehensive cross-view mappings, even in scenarios where only partial initial correspondence

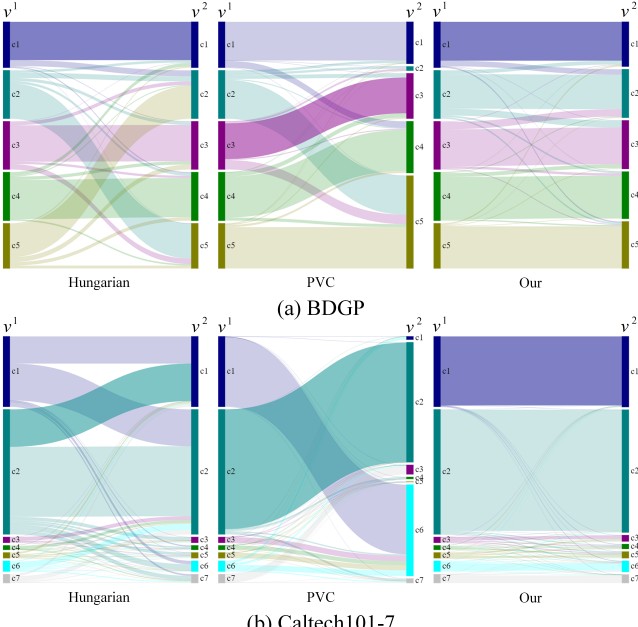

(a) BDGP

(b) Caltech101-7

**Figure 6: The Sankey diagram illustrates the view alignment obtained through the Hungarian algorithm, PVC, and our method on the BDGP and Caltech101-7 datasets. Each column on both sides represents two views, and different colors indicate different clusters. The ideal view alignment should exhibit a one-to-one mapping between the two columns with the same cluster.**

is available. Furthermore, it leverages well-learned view-alignment to exploit consistency and complementary information, thereby effectively learning meaningful semantic features. Comprehensive experiments conducted on real-world datasets demonstrate the effectiveness of our proposed approach for PVC tasks. Furthermore, a model analysis study confirms that our approach effectively addresses the challenge of learning precise view correspondence.

## Acknowledgments

This work was supported in part by National Natural Science Foundation of China (Project No.62106136, No. 62072189), in part by the GuangDong Basic and Applied Basic Research Foundation (Project No. 2022A1515010434, 2022A1515011160, 2024A1515011437), in part by TCL Science and Technology Innovation Fund (Project No. 20231752), in part by the Research Grants Council of the Hong Kong Special Administration Region (Projection No. CityU 11206622) and in part by City University of Hong Kong (Project No. 7005986).

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
