# OpenReview forum: "Contrastive Graph Distribution Alignment for Partially View-Aligned Clustering"
_acmmm.org/ACMMM/2024/Conference — MM2024 Poster_

### Official Review · Reviewer_mPL6 · 2024-05-07

**Rating:** 6
**Confidence:** 4

**Summary:**

This paper presents a contrastive graph distribution alignment for partially view-aligned clustering problem.  Partially view-aligned clustering is a challenging task as it requires a comprehensive exploration of complementary and consistent information in the presence of partial alignment of view data.  The proposed method adopts graph distribution alignment to facilitate the precise learning of view alignments.  Additionally, it incorporates cross-view contrastive learning to generate semantic features that exploit consistency information by leveraging well-established cross-view correspondence.  The experimental results demonstrate that this method achieves excellent performance for PVC tasks.

**Strengths:**

1.  This paper is clearly written and easy to follow.
2.  The full pipeline is clearly explained.
3.  Each component of the proposed method is properly designed  and seamlessly integrated into the overarching framework.  The individual modules, including graph construction, contrastive learning, and distribution alignment, operate in harmony, creating a unified and effective solution to the PVC challenge.
4.  This work is the well-adopted graph distribution matching strategy to address the challenges in the partially view-aligned clustering task.  This is a new and insightful approach that sets the work apart from existing methods.
5.  The experimental studies are also well-designed and conducted, providing a thorough evaluation of the proposed technique.  The results presented demonstrate the superior performance of the method compared to other state-of-the-art approaches.

**Limitations:**

1. The author claims that their model is capable of addressing the view-aligned problem from a graph distribution matching perspective, which is distinct from existing methods that focus on Euclidean distance. To strengthen this claim, the author should provide more experimental evidence demonstrating the superiority of adopting graph distribution matching over Euclidean distance.
2. The view-alignment module is designed to identify a permutation that minimizes the L2-Wasserstein distance between the distributions of the original graph and the alignment graph. To strengthen the understanding and credibility of this approach, it would be beneficial to include more detailed formulations of this L2-Wasserstein metric.
3. What is the origin of partially aligned data generation? Please provide a detailed explanation of the processing involved in constructing partially aligned data.

**Suitability:**

3

---

### Official Review · Reviewer_UrmU · 2024-05-20

**Rating:** 5
**Confidence:** 3

**Summary:**

The authors have introduced a contrastive graph distribution alignment approach to address the partially view-aligned clustering problem. In contrast to conventional methods that heavily rely on latent feature spaces and Euclidean distances, the authors proposed a novel perspective by utilizing similarity graphs to capture semantic-invariant structural information across views. The experimental results demonstrate the effectiveness of the proposed method for the partially view-aligned clustering.

**Strengths:**

1. Unlike the conventional reliance on latent feature spaces with Euclidean distances, the authors introduce a new perspective for the partially view-aligned clustering problem.

2. This paper is well-written and well-organized.

3. A comprehensive model analysis illustrates the detailed effectiveness of our approach. The results confirm that the proposed method effectively addresses the challenge of the partially view-aligned clustering problem.

**Limitations:**

1.While this work successfully addresses the partially view-aligned issue in multi-view clustering, it remains unclear whether the proposed method can handle the completely unaligned problem in this setting, which is also important in multi-view clustering.

2.The clustering results are obtained from semantic features with the application of K-means algorithm. I have found that the model generates similarity graphs based on semantic features. It would be beneficial by comparing to perform spectral clustering on these similarity graphs .

3.Although the paper introduces a novel approach to tackling the challenge of partially view-aligned clustering with a graph distribution-matched method, and most existing approaches address view alignment based on Euclidean distances on latent features, how about performing Euclidean distances on the similarity graph? It would be better to conduct additional experiments to further verify the effectiveness of the graph distribution-matched approach compared to employing Euclidean distances for graph matching.

**Suitability:**

2

---

### Official Review · Reviewer_tC32 · 2024-05-23

**Rating:** 5
**Confidence:** 4

**Summary:**

Paper Summary: In this paper, the authors introduced a novel approach to address the challenge of partially view-aligned clustering. The method learns view correspondence from the latent space. To evaluate the effectiveness of the proposed method, the authors conducted many experiments on several multi-view datasets under various settings. The paper is well-organized and presents sound results.

Review: The review details can be found in strength and weakness.

•  Relevance: Relevant to researchers in subareas only
•  Significance: Moderately significant
•  Novelty: Somewhat novel or somewhat incremental
•  Evaluation: Sufficient

**Strengths:**

1. This paper presents a new approach for addressing partially view-aligned clustering through the utilization of contrastive learning and graph distribution alignment. The proposed method is relatively novel and technically sound.

2. Extensive experiments show the proposed method achieve a better performance than competing method.

3. The paper is well-motivated and easy to follow.

**Limitations:**

My concerns are listed as follows:
1. The illustration in Figure 1 can be somewhat confusing, the authors have employed different symbols to denote the sample indices for each view. Specifically, for View 1, alphabetical characters are utilized to denote the sample indices, whereas numerical characters represent the indices for View 2.

2. The proposed method introduces an additional semantic feature learning module to learn similarity graph for view-alignment learning. One may question why the method doesn't directly learn the similarity graph for view-alignment learning. Providing a more detailed illustration of the motivation for the semantic feature learning would enhance the clarity of the paper.

3. The partially view-aligned clustering method aims to learn accurate view-alignment and exploit consistency information to enhance clustering based on the learned view-alignment. However, incorrect learned view-alignment may degrade clustering performance in this problem. Therefore, it is necessary to compare the performance with single-view clustering methods to assess the effectiveness of the proposed approach.

4. In Figure 7, it is unclear how the similarity graph restored using the ground truth alignment matrix is obtained and the similarity graph obtained through the learned view-alignment matrix.

**Suitability:**

3

---

### Official Review · Reviewer_jNAw · 2024-05-24

**Rating:** 5
**Confidence:** 2

**Summary:**

The paper focuses on the issue of partially view-aligned clustering where multiple views of data may not be perfectly synchronized or aligned. Traditional methods often struggle with this misalignment because they assume perfect correspondence across views. The authors propose a novel method that leverages graph distribution metrics to understand and align the semantic relationships across different views, even when these views are only partially aligned.

**Strengths:**

Innovative Approach: The paper introduces a novel methodology for aligning partially view-aligned clustering using graph distribution metrics. This approach allows for capturing semantic view-invariant instance relationships more effectively.
Improved Accuracy: By leveraging optimal transport with graph distribution, the method offers precise alignment of views, enhancing the clustering accuracy in scenarios where traditional methods based on Euclidean distances might fail.
Robustness: The use of graph-based metrics provides robustness against variations in data quality across views, which is often a limitation in multi-view clustering tasks.
Cross-view Consistency: The incorporation of contrastive learning across views helps in extracting consistent and meaningful semantic features, which are crucial for the reliability and interpretability of clustering results.

**Limitations:**

Complexity and Scalability: The complexity of the graph distribution calculations and optimal transport could pose challenges in terms of computational cost, especially when scaling to large datasets.
Dependency on Parameters: The performance of the method can be sensitive to the choice of parameters in the graph distribution and optimal transport algorithms, which may require careful tuning and validation.
Initial Alignments: The effectiveness of the method partially relies on the initial alignment of the views. Inaccuracies in these initial alignments could propagate errors through the clustering process.
Generalization Concerns: While effective in the tested scenarios, the generalization of this approach to other types of data or clustering tasks without partial view alignment remains to be thoroughly evaluated.
Specifically, there are the following：

1、The paper assumes that latent features are uniformly informative across different views. This assumption might not hold in practical scenarios where inter-view discrepancies significantly affect feature quality. A more robust approach would include mechanisms to assess and handle varying feature quality.

2、 The methodology section lacks a clear explanation of the parameters used in the graph distribution metrics and how they are optimized. Specific details on parameter settings during experiments are crucial for reproducibility and understanding the sensitivity of the model to different configuration.

3、While the paper introduces a novel approach using graph distribution metrics, it does not provide a thorough analysis of the computational efficiency. Given the complexity of graph-based methods, it is essential to discuss computational demands, especially when scaling to larger datasets.

4、The contrastive loss used for learning semantic features might not be optimal. The paper should explore alternative or additional loss functions that could potentially enhance feature learning by reducing the impact of noisy or irrelevant features.

5、The experiment section could be improved by including a broader variety of datasets with varying degrees of view misalignment. This would help validate the robustness of the proposed method across different practical scenarios.

**Suitability:**

2

---

### Meta-Review · Area_Chair_gwHe · 2024-07-01

**Recommendation:** Accept (Poster)
**Confidence:** 5

**Metareview:**

This work focuses partially view-aligned multi-view clustering. In this paper, the authors introduced a novel methodology for aligning and clustering partially view-aligned data using graph distribution metrics. The experiments are convincing. Four reviewers consistently vote to accept the paper.